# Comparing Macroscale and Microscale Walkability Indicators to Establish Pick-Up/Drop-Off Locations for a Microtransit Service in a Suburban Area

Gabriele D'Orso [1,*], Leonardo Minaudo [1,2] and Marco Migliore [1,2]

1   Department of Engineering, University of Palermo, Viale delle Scienze, Building 8, 90128 Palermo, Italy; leonardo.minaudo@unipa.it (L.M.); marco.migliore@unipa.it (M.M.)
2   National Sustainable Mobility Center (Centro Nazionale per la Mobilità Sostenibile—CNMS), Via Durando 39, 20158 Milano, Italy
*   Correspondence: gabriele.dorso@unipa.it

**Abstract:** Microtransit is a shared mobility service that operates between fixed-route transit and ride-hailing. It operates with a fleet of vans or minibuses within a service zone that is usually located in a rural or suburban car-oriented area with a transport demand that is temporally and spatially dispersed. Microtransit often expects customers to walk a short distance to pick-up/drop-off (PUDO) locations. The PUDO points need to be quickly, easily, and safely reachable by pedestrians. Thus, PUDO locations must be chosen after analyzing the walkability of the suburban area served by microtransit. This paper presents a comparison of macroscale and microscale indicators to assess the walkability of suburban neighborhoods where microtransit has to be introduced. We chose three suburban neighborhoods (Partanna Mondello, Tommaso Natale, and Mondello) in Palermo, Italy, as a study area, aiming to identify the best places to locate PUDO stops for a microtransit service. A GIS database has been built associating each link with a series of qualitative and quantitative attributes. Finally, we developed a walkability index that indicates the attractiveness of specific locations in terms of intermodal walkability. We also identified the critical pedestrian links that need actions to improve their walkability.

**Keywords:** microtransit; walkability; GIS

## 1. Introduction

The term "microtransit" refers to a particular type of demand-responsive transit (DRT), i.e., on-demand transport systems, in which fleets of vehicles are used to pick up and drop off users at places they request, depending on their needs, acting as a service that fills the gap between individual transport (cars) and conventional public transport (bus lines) [1,2]. Microtransit is characterized by the use of more compact vehicles [3], such as vans or minibuses, which make it possible to extend the coverage of the service to all those areas with inaccessible infrastructures for conventional buses. This type of service is also particularly effective in less densely populated areas with a low transport demand that is spatially and temporally dispersed, such as rural or suburban areas, where conventional public transport lines are not able to properly meet the users' needs [1]. Thus, it could be designed as a first-mile/last-mile feeder to a fixed route service, as a filler for gaps in fixed route service, or as a replacement for a portion of low-ridership fixed routes in low-density areas. These on-demand services rely on digital platforms, which allow users to book a ride by choosing the starting point and the destination of their trips [4], thus better meeting the existing travel demand. Although they are flexible services, looking at the design of schedules and routes, they can take on several configurations characterized by different levels of flexibility [1]: the spectrum for DRT services goes from services with fixed routes and fixed schedules but the ability to book a seat and to request adjusted

routes, like employee shuttles and school busses, to ones with fixed routes and on-demand schedules, with the ability to make detours and operate with physical stops, virtual stops, curb-to-curb or corner-to-corner, and finally to fully flexible (in time and locations) services with door-to-door options.

Microtransit is an affordable alternative to owning a car for individuals that live in hard-to-serve areas. This service aims to enhance accessibility to amenities and connect to the main transportation hubs, guaranteeing a higher reliability than conventional public transportation services, especially during off-peak times (i.e., at night).

Considering a stop-based microtransit service, where users can book a ride by selecting some predefined stops as pick-up and drop-off points (PUDO), it seems clear that finding the optimal locations for PUDOs is fundamental, since they must be as accessible as possible. The identification of locations for PUDOs should be one of the first steps in designing a stop-based microtransit service. Indeed, once the ride has been booked by the users, the service management system ensures that one of the vehicles in the fleet arrives at the indicated stops in order to fulfil the trip requests, while users have to walk to access the service.

To adequately extend service coverage, the stops must be easily accessible by walking by the majority of the population residing in the area to be served. Furthermore, elderly people and people with disabilities appear to be among the users for whom the service is especially attractive [5,6], thanks to the reduced waiting times and greater coverage compared to a conventional public transport service. Indeed, microtransit does not require users to walk long distances and wait a long time at a bus stop, which is a stated reason especially for older travelers to reject public transport [7]. Thus, it is essential to assess the walkability of the area where the service is to be implemented.

By comparing macro-scale and micro-scale walkability indicators in assessing the best, most walkable locations for microtransit stops, the paper aims to develop a methodology for microtransit operators to locate PUDOs along microtransit routes and for policymakers to identify priority actions to improve the walkability of these locations.

We chose the northern suburbs of the city of Palermo, Italy, as a study area, focusing on the districts of Partanna Mondello, Tommaso Natale and Addaura. We initially assessed the walkability of the study area following a macro-scale approach, using well-established indicators in the literature such as road density, intersection density and pedestrian catchment area. QGIS software and its plugins were used to map the indicators; this made it possible to identify the most suitable locations for the stops along pre-defined microtransit routes in the study area. However, we wanted to verify that the locations identified using a macro-scale walkability assessment were equally walkable considering the quality of the walkways.

Thus, we used a micro-scale approach to consider the local characteristics of the single walkways, assessing several indicators related to the quality of the pedestrian realm.

The remainder of the paper is structured as follows. Section 2 gives a brief overview of the methodologies for assessing walkability. We describe the macro- and micro-scale walkability assessments for finding the locations of microtransit stops in Section 3, presenting a case study. Then, the results are discussed in Section 4. In Section 5, we discuss how the combination of both the approaches is useful to design microtransit services for suburban areas, and how to improve the walkability assessment in further studies.

## 2. Background

Walkability refers to the ability to walk comfortably and safely within urban areas, especially to reach the main points of interest in the city, such as railway stations, public transport stops, schools, shops, and parks. Encouraging pedestrian mobility is of primary importance, as it has zero impact in terms of pollutant emissions and is associated with important benefits for human health, reducing the risks of cardiovascular diseases and obesity [8]. Having good-quality pedestrian infrastructures could therefore encourage people to walk to satisfy their daily needs. In particular, increasing the quality of the pedestrian routes around bus stops and railway stations could increase the attractiveness of public

transport and favor walk-to-transit, as users are guaranteed the possibility of comfortably and safely walking the first/last mile [9–11]. A pedestrian-friendly environment is a key component for Transit-Oriented Development [12] and a key step towards the achievement of UN sustainability goals 3 and 11.

A first approach to the walkability assessment can be made at the macro scale, by carrying out an assessment at the city or neighborhood level [13]. In the macro-scale approach, walkability is related to pedestrian accessibility, mainly in terms of network connectivity.

Many macro-scale walkability indicators have been reported in the literature, and researchers have developed several tools to assess walkability at a city or neighborhood level, combining these indicators together.

Block length, maximum block size or block density, i.e., the mean number of blocks per square kilometer, have been used to measure connectivity and estimate the walkability of an urban area.

Other connectivity measures like intersection density, i.e., the number of road intersections per square kilometer, and road density, i.e., the sum of road lengths per square kilometer, were found to be very important in encouraging walking when holding high values [14]. A higher number of intersections means more path choices for the users. Moreover, very large blocks with few pedestrian crossings are less suitable for pedestrian mobility than neighborhoods with a denser street network, with more small blocks and frequent crossings.

Another common indicator is the pedestrian catchment area (PCA) ratio [15,16], also known as the Ped-Shed ratio, evaluated as the ratio between the network-defined pedestrian service area within a given walking distance (i.e., 500 m) from a location and the theoretical pedestrian service area, represented by a circle of that radius (i.e., 500 m) around the same point. Thus, the first area is calculated considering the network distances, while the second one only considers the Euclidean distances (i.e., "as the crow flies").

The scientific literature on macro-scale walkability usually also considers some indicators describing the land use mix and population density [17] as main walkability indicators, as well as the numbers of amenities within a walkable distance. For example, Walk Score [18] is a walkability measure related to some macroscale indicators such as the types and numbers of provided amenities, catchment distances to these amenities, neighborhood block lengths, and the density of street intersections.

Also, the number of people residing and the number of employees working in the catchment area of a bus stop or a railway station are frequently used to express the walkability of the area in terms of pedestrian accessibility [19–21].

However, microtransit services usually operate in rural or suburban areas, which are characterized by a low population density and mainly residential use, with the presence of small local shops and few commercial or industrial spots; therefore, while these kinds of indicators can be relevant in an urban context, they may not be that useful in our case. Indeed, as stated in [22], walkability indicators in rural and urban areas may have different degrees of relevance. We identified intersection density, road density and PCA ratio as the more appropriate indicators to proceed to a macro-scale walkability assessment, aiming to find the best locations for microtransit PUDOs in a suburban area.

The second (micro-scale) approach, on the other hand, pays greater attention to the actual conditions of pedestrian routes, analyzing their quantitative and qualitative characteristics with an impact on their attractiveness [13], and investigating a more or less extensive range of factors that may be objective or linked to the pedestrian's perception [23].

In this walkability assessment, various factors are considered, which generally focus on how passable, safe, and pleasant to walk the walkways are [24]. Many researchers have developed micro-scale walkability indices by combining different factors. For example, Bartzokas-Tsiompras et al. (2023) [25] used indicators grouped by three dimensions: the sidewalk environment dimension, including the presence of street lights, the maintenance of sidewalks, and the presence of trees; the pedestrian crossings dimension, including the presence of walk signals, curb ramps, and marked crosswalks; and the streetscape-level

dimension, including the presence of well-maintained buildings and the absence of graffiti. On the other hand, D'Alessandro et al. (2016) [26] made a micro-scale walkability assessment based on 12 indicators gathered into four categories: practicability (sidewalk surface, obstacles, road slope), safety (protection from vehicles, road lighting, crossing protection), urbanity (sidewalk width, road equipment, land mix) and pleasantness (vehicular traffic, building context, green space).

Although some pedestrian routes have good values as regards macro-scale walkability indicators, they may be characterized by critical aspects such as, for example, route discontinuity due to the presence of obstacles, the absence of pedestrian-oriented lighting, or the bad condition of the sidewalk, which reduce the actual attractiveness of the route [24].

## 3. Methodology

We compared the macro-scale and micro-scale walkability assessments through the following steps:

- GIS mapping of macro-scale indicators within the study area;
- Finding the stops' locations along the microtransit routes considering the macro-indicators (overlay analysis);
- GIS mapping of the micro-scale indicators around the identified stops and assessment of a quality index;
- Identification of the most traveled pedestrian routes and definition of an intervention priority index.

As will be discussed in Section 5, these steps could represent the steps of a combined methodology for assessing the walkability of the areas around microtransit stops. The methodology framework is reported in Figure 1.

A suburban area in the northern part of Palermo, Italy was chosen as the study area. It is a predominantly residential area with few points of interest, including a shopping center, restaurants, and hotels. The area is characterized by low travel demand; it is spatially and temporally dispersed, and served by low-quality public transport services—this makes microtransit a viable option for improving the accessibility and quality of public transport in the study area. Moreover, a previous study conducted in Palermo [27] showed that the time spent walking to stops is more bothersome than longer in-vehicle times, from citizens' perspectives. Therefore, introducing a transport service able to reduce walking times to stops is a significant way to improve the attractiveness of the public transport in the study area.

The microtransit routes were defined based on previous studies [28]. Indeed, an SP survey was conducted in the study area, and a four-step model was developed to estimate the microtransit demand. The microtransit network was built considering public roads with a sufficient width for allowing vehicles to stop safely and without hampering traffic. Then, we performed an O/D matrix assignment, considering the estimated microtransit trips and identifying the microtransit routes where the highest flows were assigned. The resulting routes consist of a fixed line and several detours that are traveled only once a trip request occurs.

### 3.1. Mapping of Macro-Walkability Indicators

As mentioned, microtransit uses small-scale vehicles, able to travel along roads that would be unpassable for ordinary vehicles. This results in a capillary service, allowing users to reach the service's stops by short walks. Therefore, it is essential to ensure high accessibility to stops, so that these can be reached by many people as possible. Accessibility, in fact, is an essential element for the walkability of a given area [29,30]. Macro-scale walkability indicators related to accessibility have to be assessed and the stops can be placed in locations with the best values. We considered road density, intersection density and PCA as indicators to find the best locations along the microtransit routes in suburban areas. Considering several indicators allowed us to overcome the limitations of any single one; indeed, PCA does not allow us to appreciate how interconnected the network is or

how large or uniform the individual blocks of the urban fabric are [15], which is instead measured by the other indicators.

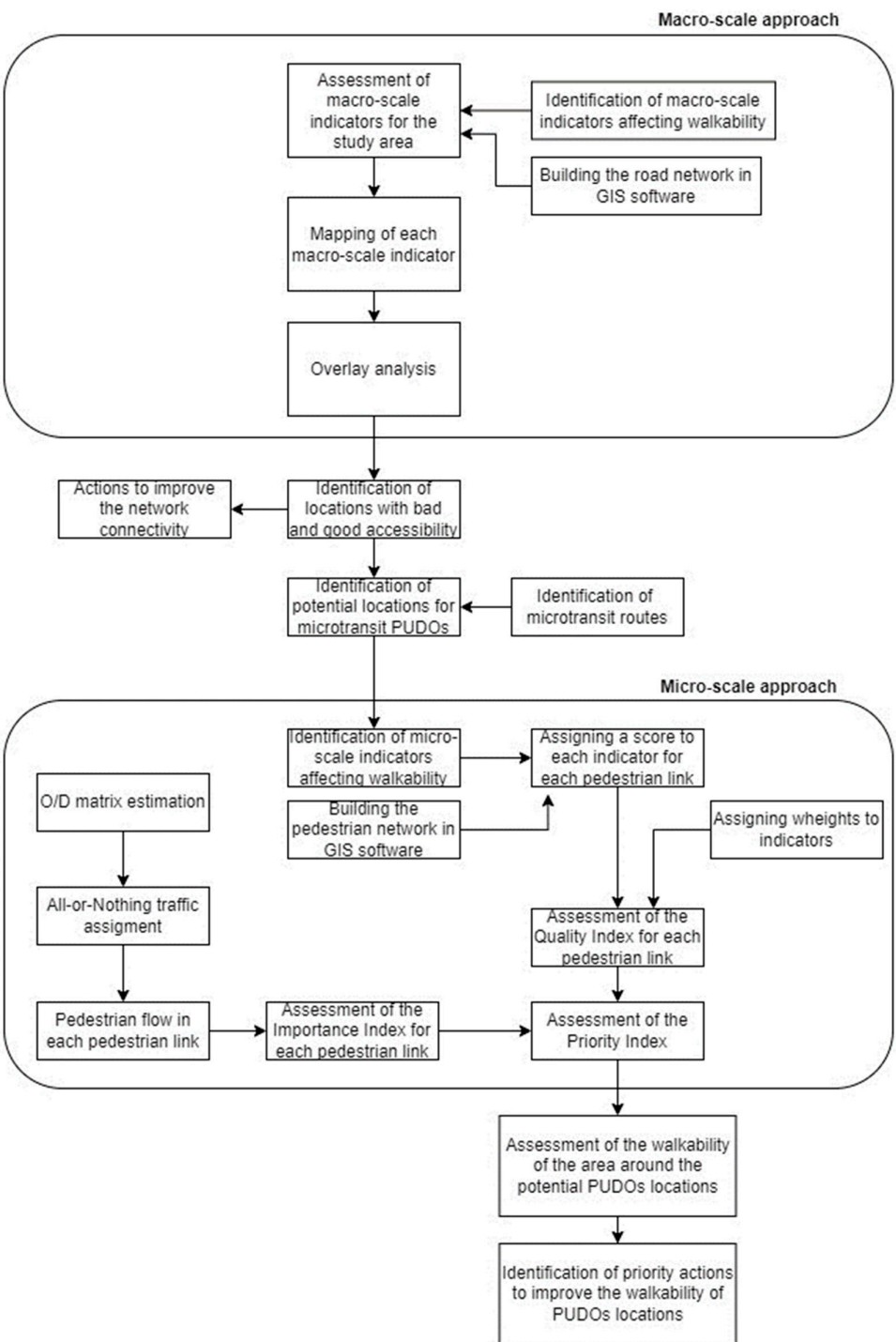

**Figure 1.** Methodology framework.

The evaluation of the macro-indicators was carried out using QGIS software (version 3.28.2). The road network of the study area was imported from OpenStreetMap using the QGIS plugin "QuickOSM". The road graph was built using the "Networks" plugin, resulting in a layer containing the roads as links and a layer containing all the road intersections and dead ends as nodes.

Road density was evaluated using the tool "Line Density", which generates a raster whose cells are assigned a density value given by the sum of the lengths of the linear

elements falling within a circular surrounding divided by its area. As input to the command, a surrounding area equal to 1 sq. km was generated. Therefore, each cell of the raster represents the point value of the road density.

With regard to the intersection density, a layer was created containing a 100 m × 100 m grid of points, and then a buffer of 1 sq. km was created around each point. Finally, the number of road intersections within the buffer was counted for each point in the grid using the "Count points in polygon" tool, which returns the number of points within a polygon. To obtain the raster showing the spatial variability of the parameter within the study area, a TIN interpolation was performed.

The pedestrian catchment area ratio was evaluated from the points representing the road intersections. Considering 500 m as the network distance people are willing to walk [17], PCA was calculated for each road intersection as the ratio between the actual reachable catchment area within 500 m from the point representing the road intersection via the road network and the unimpeded reachable area represented by a 500 m Euclidean buffer centered in the same point, as shown in Equation (1). The buffers were created using the "Buffer" tool, while the actual reachable area was evaluated using "Area to be Served" and "Convex Polygons" tools.

$$\text{PCA} = \frac{\text{Actual Reachable Area}}{\text{Buffer Area}} \tag{1}$$

Also, in this case, once the parameter for the various points had been calculated, a map was produced using the TIN interpolation function.

Figure 2 shows the spatial variability of road density (a), intersection density (b), PCA (c) and the result of an overlay analysis (d). Considering the predefined microtransit routes, the most walkable location to place a microtransit stop is in Via Iandolino (red dot in Figure 2d), presenting the highest value for the overlay of the three macro-indicators (overlay—0.84; road density—14.5 km/km$^2$; intersection density—103 int./km$^2$; PCA—0.6). We considered only this stop to proceed with the micro-scale approach.

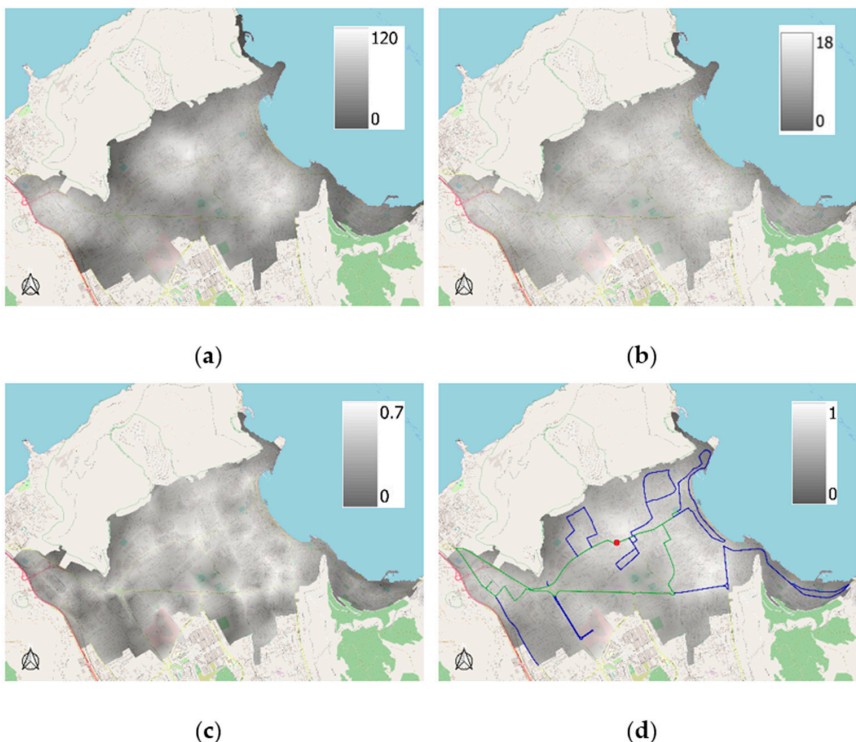

**Figure 2.** (**a**) Intersection density; (**b**) road density; (**c**) pedestrian catchment area; (**d**) overlay analysis and microtransit routes (green line represents the fixed line, blue lines represent on-demand detours).

### 3.2. Mapping of Micro-Walkability Indicators

Considering the walkable area around the identified stop located in Via Iandolino, a micro-scale approach was followed. We used the micro-walkability assessment tool previously developed by D'Orso and Migliore (2020) [24], but differentiating indicators for sidewalks and crosswalks.

First, the pedestrian network was traced using QGIS. The level of safety, practicability, and pleasantness of each pedestrian link (sidewalk or crosswalk) was assessed, considering a set of indicators (Tables 1 and 2), through site inspections and using Google Street View. Finally, a "Quality Index" was associated to each pedestrian link.

**Table 1.** Micro-walkability indicators for sidewalks.

| Category | Weight | Indicator | Value | Description |
|---|---|---|---|---|
| Practicability | 0.3 | Pavement slope | 0 | Steep slope for elderly and wheelchair users (>5%) |
| | | | 1 | Non-perceptible slope (<5%) |
| | | Pedestrian LOS | 0 | High pedestrian flows, insufficient sidewalk width or presence of obstacles (LOS = E, F) |
| | | | 1 | LOS = C, D |
| | | | 2 | Low pedestrian flows, adequate pavement width or absence of obstacles (LOS = A, B) |
| | | Decay | 0 | Very degraded pavement |
| | | | 1 | Presence of some potholes or depressions |
| | | | 2 | No potholes or depressions, pavement in good condition |
| Pleasantness | 0.3 | Street furniture | 0 | Absence of litter bins, benches, and other elements of street furniture |
| | | | 1 | Presence of litter bins, benches, and other elements of street furniture |
| | | Sun/Rain protection | 0 | Lack of shelter from the sun or rain |
| | | | 1 | Presence of shelters from sun or rain |
| | | Green areas | 0 | Absence of flowerbeds or green areas |
| | | | 1 | Presence of flowerbeds or green areas |
| | | Shops | 0 | Absence of shops |
| | | | 1 | Presence of shop windows and shops |
| | | Architectural context | 0 | Degraded architectural context |
| | | | 1 | Pleasant architectural context |
| Safety | 0.4 | Lighting | 0 | Poor or no lighting |
| | | | 1 | Adequate and efficient street lighting according to UNI standard (UNI11248) |
| | | Traffic flow and speed | 0 | High traffic volumes (>1000 vph) or high speeds (>50 km/h) |
| | | | 1 | In other cases |
| | | | 2 | Free flow (<300 vph) and low speeds (<30 km/h) |
| | | Protection from vehicles | 0 | Absence of barriers to protect pedestrians from vehicles |
| | | | 1 | Presence of protection elements |
| | | Driveways | 0 | Presence of driveways along the route |
| | | | 1 | Absence of one or more driveways |

**Table 2.** Micro-walkability indicators for crosswalks.

| Category | Indicator | Value | Description |
|---|---|---|---|
| Safety | Lighting | 0 | Poor or no lighting |
| | | 1 | Pedestrian-oriented lighting |
| | Traffic flow and speed | 0 | High traffic volumes (>1000 vph) or high speeds (>50 km/h) |
| | | 1 | In other cases |
| | | 2 | Free flow (<300 vph) and low speeds (<30 km/h) |
| | Traffic lights | 0 | Absence of traffic lights at the intersection |
| | | 1 | Traffic lights at the intersection but presence of conflicts between different vehicle components |
| | | 2 | Traffic lights that eliminate conflict points between vehicles and pedestrians |
| | Visibility | 0 | No crosswalks or very poor crosswalk visibility |
| | | 1 | Good crosswalk visibility |
| | | 2 | Excellent crosswalk visibility |

The quality index (QI) was calculated using Equation (2) for sidewalks and Equation (3) for crosswalks, whereby the categories were replaced by the sum of the scores obtained in the associated micro-indicators. Each link can obtain a score between 0 and 5.

In Equation (2), the weights for practicability, pleasantness and safety are 0.3, 0.3 and 0.4, respectively. These weights derive from a study involving the city of Palermo [24]. The results of this study show that safety has a bigger impact on walkability for users. Since the maximum reachable score of the safety category for crosswalks is 7, we considered a coefficient equal to 5/7 in Equation (3) to obtain a quality index between 0 and 5 also for crosswalks.

$$QI_{sw} = \text{Praticability} \times 0.3 + \text{Pleasantness} \times 0.3 + \text{Safety} \times 0.4 \tag{2}$$

$$QI_{cw} = \text{Safety} \times \frac{5}{7} \tag{3}$$

Figure 3 shows the values that QI assumes in each link of the area around the micro-transit stop. It can be noted that, although the stop location has good accessibility according to the macro-scale approach, some road segments present a very low quality.

### 3.3. All or Nothing Assignment

Once noted that some walkways around the microtransit stop have a low quality in terms of practicability, pleasantness, and safety, we must understand if there is the urgent need to improve some of them. For this purpose, we must assess which roads would be the most traveled by microtransit users to reach the stop, and also by pedestrians to reach the amenities present in the area.

To assess the pedestrian flows in each pedestrian link of the Iandolino area, an O/D matrix including daily non-commuting walking trips and microtransit trips made by people over 14 years was assigned onto the pedestrian network. The whole study area was divided into Traffic Analysis Zones (TAZs) and centroids were assigned to them. However, the demand originating in the TAZs around the Iandolino stop was distributed to the residential buildings proportionally to their volume. Thus, the residential buildings of the Iandolino area were considered as potential origins, while points of interest such as grocery shops, postal offices and banks were considered as destinations. Also, centroids of the TAZs outside the Iandolino area were considered as origins or destinations for walking trips. Moreover, the microtransit stop was considered as an origin for the walking

part of the microtransit trips starting from the other parts of the study area and having destinations inside the Iandolino area, or as a destination for microtransit trips starting from the residential buildings of the Iandolino area and having a destination in the other TAZs. The matrix predicts a total of 3652 pedestrian trips per day.

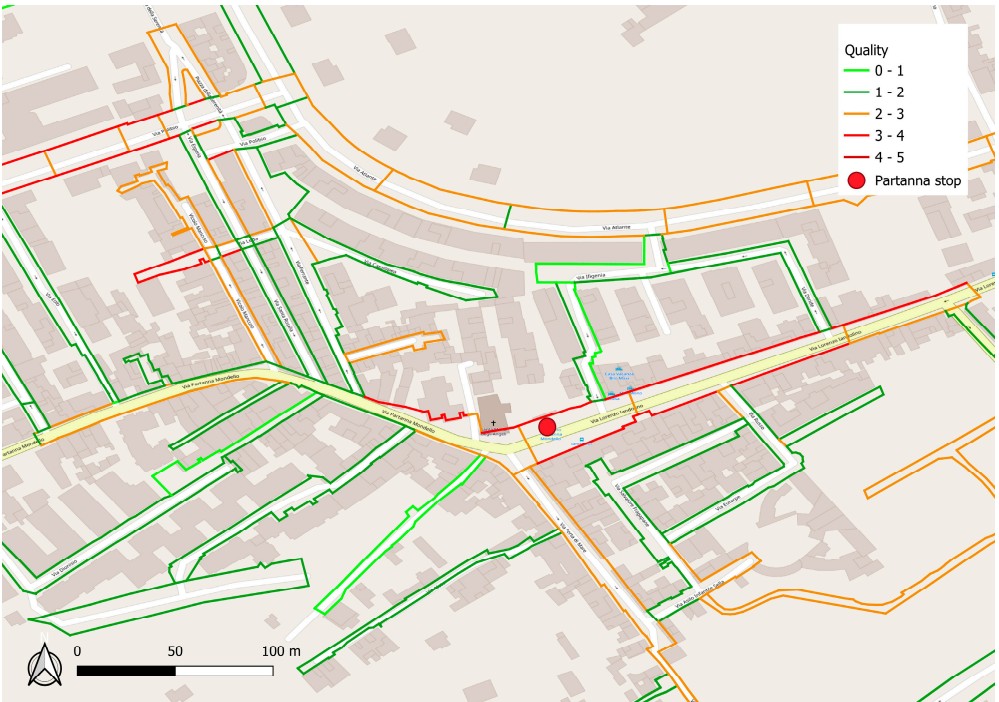

**Figure 3.** Quality map of walkways.

An All or Nothing assignment was carried out using "AequilibraE" (version 0.6.2), an open source QGIS plugin, which, for each O/D pair, assigns flows to the shortest route, regardless of traffic conditions and quality. We decided to use the 3.10.14 version of QGIS to properly use the plugin without bugs, which we have found in newer versions. The choice of this type of assignment lies in the need to identify the routes that users would choose under ideal conditions, thus without considering the actual quality level. Indeed, other things being equal, users choose the shortest path to reach their destinations. After the assignment was performed, we checked the results in order to understand if the algorithm of the software was assigning the demand to the shortest paths properly, and whether the pedestrian network was created correctly (e.g., presence of topological errors like undershoots, overshoots, and dangling nodes).

The highest flow was obtained in correspondence with the stop, with 752 pedestrians per day (Figure 4). The network links were ranked on a scale of importance from 0 to 5 according to the flow assigned to them. A value equal to 0 identifies a link with flow under 100 pedestrians per day, while, continuing in steps of 100, a value equal to 5 means having a pedestrian flow of more than 500 pedestrians per day.

### 3.4. Priority Identification

The assessment of the quality of the routes and the calculation of the potential flow values on each link made it possible to prioritize the maintenance of pedestrian routes, giving greater priority to those links with greater flows and lower quality.

As mentioned, in fact, the flows assigned by All or Nothing represent the user's desire paths without taking into account the real conditions in terms of their quality, but, in reality, users may be led to choose a different, and therefore longer, path due to the poor conditions of the ideal path identified by the assignment [30]. This clearly compromises the experience of users who choose to walk to the microtransit service stop.

Thus, the following equation was used to form a priority index (PI):

$$PI = Importance - QI + 6 \qquad (4)$$

The priority index has a range from 1 to 11, where 11 stands for the highest intervention priority. Figure 5 shows the priority map.

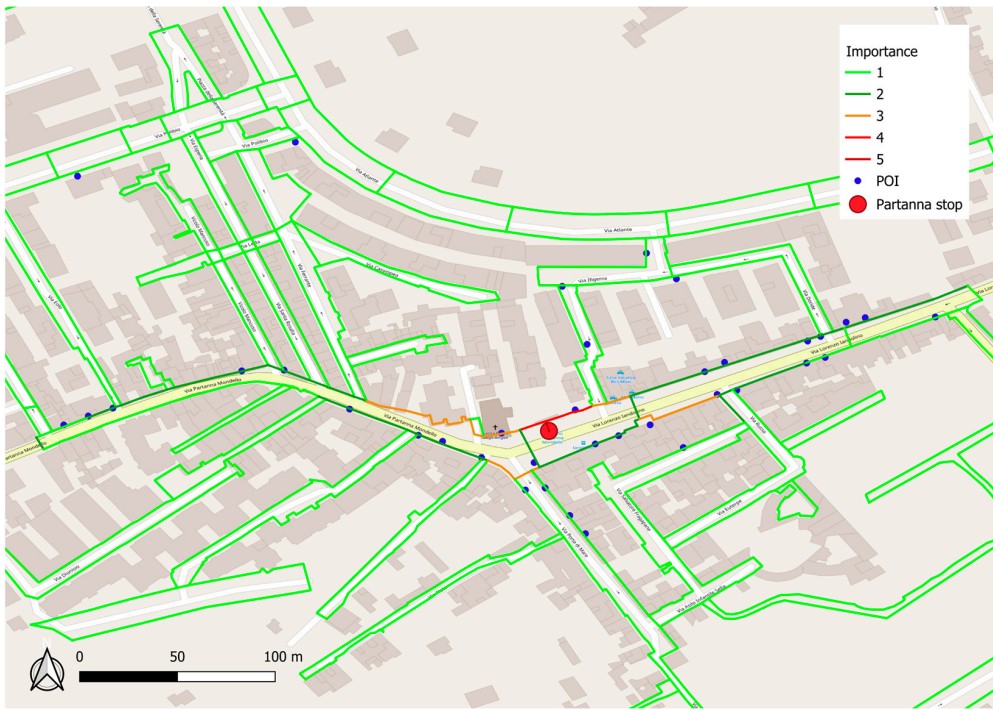

**Figure 4.** Importance of walkways in terms of pedestrian flows.

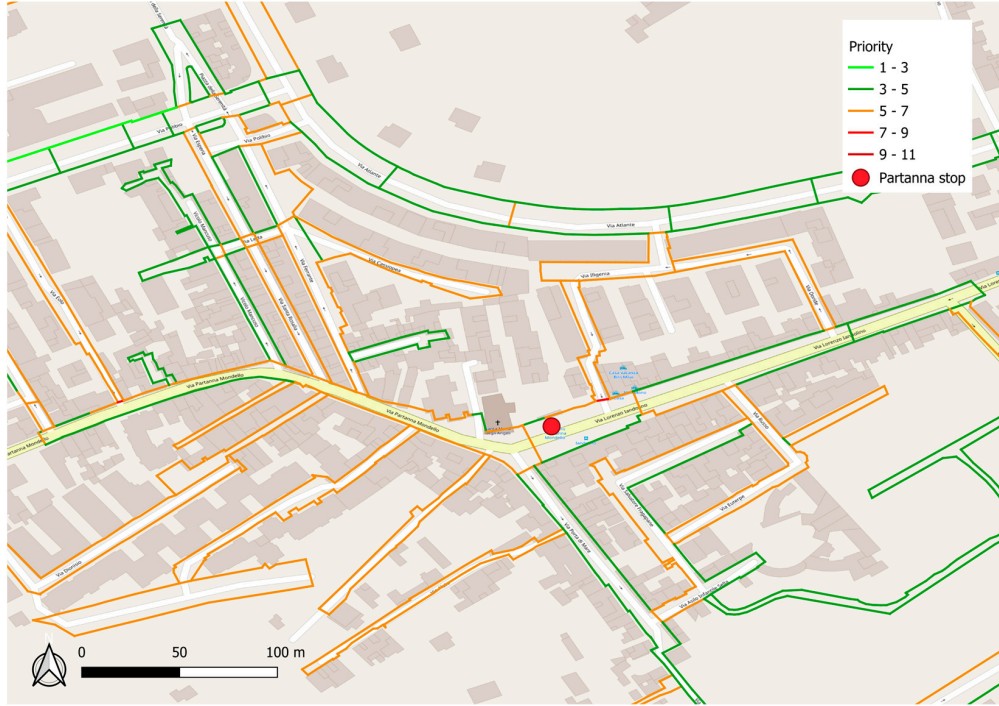

**Figure 5.** Priority map.

## 4. Results and Discussions

Considering the case study in Palermo, we compared macro-scale and micro-scale walkability assessment methods to understand the differences between these two approaches in finding the best locations for microtransit PUDOs.

The overlay analysis at the end of the macro-scale approach led to the identification of the most walkable location as a place located in via Iandolino. However, as Figure 5 shows, most of the walkways around the Iandolino stop (the orange and red ones) need priority actions to improve their walkability. In particular, the crosswalks near the stop have a low quality index, as reported in Figure 3, meaning that they could be not perceived as safe by pedestrians.

The comparison of the two approaches led to the identification of their strengths and limitations (Table 3). Using the macro-scale approach is essential to better identify how interconnected the network is, and therefore the actual possibility of pedestrians going from a certain origin to a destination without being forced to overextend their route. Therefore, the macro-scale approach can represent a first step towards identify the stops' locations, since it allows us to identify the areas characterized by the highest level of road network connectivity. On the other hand, the main limit of this approach is that it does not take into consideration the actual quality of the routes. Indeed, although a certain location can be characterized by the highest values for macro-scale indicators, it could be a not very walkable location, having unpassable or unsafe walkways around it. Therefore, the micro-scale walkability assessment can represent a more detailed approach, offering a good way to evaluate the actual quality of pedestrian links around the identified stops.

**Table 3.** Comparison between the macro-scale and micro-scale approaches.

| Walkability Assessment Approach to Locate PUDOs | Macro-Scale | Micro-Scale |
|---|---|---|
| **Scale** | City or neighborhood level | Single pedestrian route |
| **Evaluated features** | Linked to network connectivity and accessibility | Linkage to pedestrian route quality |
| **Objective** | Useful to understand places with the highest network connectivity to locate microtransit stops | Useful to investigate the quality of the pedestrian routes around the identified stops |
| **Strengths** | Easy to perform via GIS software even for large areas | More detailed approach; the identification of features to be improved could lead to short-term interventions |
| **Limitations** | It does not take into account the quality of pedestrian routes | Time-consuming due to evaluation of indicators |

However, a major limitation of the micro-scale walkability assessment is that the evaluation of microscale indicators for all pedestrian links and the evaluation of the pedestrian O/D matrix can be expensive and time-consuming depending on the number of microtransit stops to be identified and the extent of the area, while the evaluation of the macro-scale indicators using the identified methodology is quite a fast process.

The expensive and time-consuming process of walkability assessments using micro-scale indicators was also found to be problematic by Koo et al. (2022) [31], who stated that micro-scale factors have been rarely incorporated into widely used walkability indices because their measurements heavily rely on on-site surveys. However, many studies examined the possibility of replicating in-person audits with virtual audits using street view image services such as Google Street View, reducing the time required to perform the walkability assessment [32,33].

Finally, identifying micro-scale streetscape features to be improved can be particularly important because they may moderate the effects of macroscale accessibility factors [34]. Moreover, micro-scale streetscape features (e.g., presence of benches or adequate lighting) can be more easily modifiable in the short term than macroscale accessibility factors (e.g., intersection density).

## 5. Conclusions

Microtransit provides a new opportunity to improve mobility in low-density suburban areas, where traditional public transport services are often infrequent, unreliable, and suffer from a lack of access. On-demand minibuses can expand access to the main transportation hubs, job, education, and social opportunities, especially for elderly people and disabled people, representing a transportation option that will help to build more inclusive neighborhoods. However, considering a stop-based service, pedestrian accessibility to microtransit stops must be guaranteed, since walking is generally the most common transport mode used to access and egress stops.

Thus, the contribution of our paper is in demonstrating the importance of an adequate study of the walkability of an area where a microtransit service will operate. Suburban or rural areas are characterized by a road network that is not always meant to meet pedestrians' needs [35]. It is therefore essential to guarantee the ability of users to reach—safely and comfortably—all the PUDOs of a stop-based mictrotransit service. To do so, we used two different approaches in assessing walkability: first, a macro-scale walkability assessment aiming to identify the locations with the highest network connectivity and that were, for this reason, easily reachable by users; subsequently, a micro-scale walkability assessment useful to verify whether the locations identified by the previous approach were actually walkable, taking into account indicators related to the quality and safety of pedestrian infrastructures. Finally, we found that the first approach is limited, because some locations with good network connectivity can be surrounded by pedestrian infrastructures with safety issues or that are perceived as not really walkable. However, the second approach could be very time-consuming if extended to a large area. Thus, the results suggest the need to use both the macro-scale and micro-scale walkability assessments in combination with each other, and not only one of them, as also found by [22,32], in order to derive a more comprehensive understanding of how the walkability of PUDOs' locations can affect the access to microtransit by users.

Indeed, the identified methodology allows microtransit operators to better place the service stops in the area where the service will be activated, also allowing policymakers to understand which walkability features the pedestrian links around the stops lack, and to establish the intervention priorities required to improve the overall walkability of the area, as well as the accessibility of the microtransit service. In this regard, the methodology described in this paper could support decision-making, and GIS software could offer a valid and user-friendly tool to perform this kind of analysis.

The main limit of this study is that we only considered one single set of indicators, without considering the different perceptions of the indicators by different social groups, such as the elderly, people with disabilities, low-income people, or women. Further studies can be conducted to understand the changes in perceptions of groups of people characterized by different ages, genders, or disabilities. The micro-scale assessment could be done using different weights for the indicators, as well as indicators that consider the specific needs or characteristics of each social group.

**Author Contributions:** Conceptualization, M.M.; methodology, G.D., L.M. and M.M.; software, L.M.; validation, M.M.; formal analysis, G.D. and L.M.; investigation, G.D. and L.M.; resources, G.D. and L.M.; data curation, G.D. and L.M.; writing—original draft preparation, L.M.; writing—review and editing, G.D. and M.M.; visualization, G.D. and L.M.; supervision, M.M.; project administration, M.M.; funding acquisition, M.M. All authors have read and agreed to the published version of the manuscript.

**Funding:** This research has been partially supported by the European Union—Next-Generation EU—National Sustainable Mobility Center CN00000023, Italian Ministry of University and Research Decree n. 1033-17/06/2022, Spoke 9, CUP B73C22000760001. This research was also funded by PRIN of the Italian Ministry of University and Research, within the research project n. 20174ARRHT, "WEAKI TRANSIT: WEAK-demand areas Innovative TRANsport Shared services for Italian Towns", and by the European Union—FESR or FSE, National Operational Programme (NOP) on Research and Innovation 2014–2020—DM 1062/2021.

**Data Availability Statement:** The data presented in this study are available on request from the corresponding author.

**Conflicts of Interest:** The authors declare no conflict of interest.

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
