# Peer review of "Comparing Macroscale and Microscale Walkability Indicators to Establish Pick-Up/Drop-Off Locations for a Microtransit Service in a Suburban Area"

_infrastructures, doi:10.3390/infrastructures8120165_

Round 1
Reviewer 1 Report
Comments and Suggestions for Authors
The study provides a comparative analysis of macroscale and microscale indicators for evaluating the walkability of suburban neighborhoods in anticipation of the introduction of microtransit services. Three suburban neighborhoods were selected, namely Partanna Mondello, Tommaso Natale, and Mondello, located in Palermo, Italy, as our research area. The objective was to identify optimal locations for placing Pick-Up/Drop-Off (PUDO) stops for a microtransit service. However, It would be more suitable for publication if the following revisions were made.
1. It would be better that add a flowchart of the methodology.
2. Line 140 “Road Density was evaluated using the command 'Line Density',...”. Why does the Line Density use single quotes?
3. It is recommended that the “Category”, “Weight”, and “Indicator” columns of Table 1 would be vertically centered, and the same for Table 2.
4. A period is missing from the title of table 1.
5. The title of Table 1 should be “Micro-walkability indicators for sideways” to correspond to Equation 1.
6. Please specify the limitations and the contributions of the study at the end of Results and conclusions.
Reviewer 2 Report
Comments and Suggestions for Authors
MDPI review:
(1) A lack of a comprehensive literature review. Thus, the state-of-the-art research progress in the area is unknown. It would be better to survey relative studies on the selection of the macroscopic indicator, as well as the microscopic ones.
(2) L151 The pedestrian catchment area and the definition is not quite clear. It would be better if the mathematical formula could be expressed in an explicit way.
(3) L177 It is not clear to readers why the weights for practicability, pleasantness and safety are set at 0.3:0.3:0.4. The coefficient of 5/7 in Equation 2 is not justified as well.
(4) L207, the route assignment function (using QGIS’s AequilibraE’) is not convincing. Thus, it is questionable to the pedestrian flows allocated to each route and link.
(5) The interpretation of the comparison results are weak. The authors may consider using a table or some graphs to compare the differences between the two approaches, namely the macroscopic and the microscopic matrix.
Comments on the Quality of English LanguageNo comment
Reviewer 3 Report
Comments and Suggestions for Authors
The article is focused on the macroscale and microscale walkability indicators applied to microtransit service 3 in a suburban area. The issue discussed in the paper is attractive. However, the article is too short and poor (e.g. introduction and results description).
Consequently, relevant modifications are recommended for enhancing this work following a major revision.
1. Improve the introduction section. The authors must improve the literature analysis (walkability indicators, planning and methods).
2. Figures 2,3,4 in the paper are not legible. Please increase their quality.
3. Based on the usual structure of a scientific paper, the authors have to provide a result paragraph with in-depth considerations and compare their results with the literature. Consequently, the authors have to split the last paragraph into two (results and conclusion section).
Round 2
Reviewer 3 Report
Comments and Suggestions for Authors
The authors improved the manuscript therefore it can be published.